# A method to measure molecular hybridization

**Sophia Rosencrantz[1,2], Vitali Matyash[3], Ruben R. Rosencrantz[1,3,4], Oleh Fedorych[3]***

1 Fraunhofer Institute for Applied Polymer Research IAP, Life Science and Bioprocesses, Potsdam, Germany, 2 Fraunhofer Cluster of Excellence Immune-Mediated Diseases CIMD, Frankfurt am Main, Germany, 3 ICHORtec GmbH, Berlin, Germany, 4 Brandenburg University of Technology BTU, Institute for Materials Chemistry, Chair of Biofunctional Polymermaterials, Senftenberg, Germany

* o.fedorych@ichortec.com

**Data Availability Statement:** This manuscript's minimal data set is publicly available via figshare, under DOI 10.6084/m9.figshare.26206580. The direct link is: https://figshare.com/articles/dataset/PLOS_zip/26206580.

## Abstract

Fluorescence-based oligonucleotide probes have a great importance in research of molecular interactions. Molecular beacons (MBs) are special case of fluorescent probes that form a stem-loop shape, bringing together a fluorophore and quencher, thus emitting fluorescence only when hybridized to a complementary target. Here we describe a new method for the quantitation of MB hybridization based on the measurement of changes in free energy instead of the fluorescence intensity. The MB energy state can be measured by micro-fluorescence detection. The approach allowed to determine hybridization energy of the MB with target nucleotide directly from fluorescence spectra and distinguish the MB in unfolded and hybridized states. Moreover, the method enabled us to discriminate between DNA duplexes with perfect complementarity or a single-nucleotide mismatch, based on the first direct experimental prove of enthalpy-entropy compensation.

## Introduction

Nucleic acids hybridization is a powerful technique widely used for the detection of complementary sequences [1]. It involves formation of stable hydrogen bonds between complementary base pairs of a probe and a target sequence. Molecular beacons (MBs) are a class of probes in which the linear probe sequence is flanked by short, self-complementary tails that cause the free MB to form a stem-loop (hairpin) structure in solution [2]. MBs are labeled with a fluorophore at one end but carry a quencher at the other, such that the fluorescence signal is suppressed when the two are brought together in the stem of the hairpin. When the MB hybridizes to its target and forms a linear DNA duplex, the fluorophore and quencher are separated and a fluorescence signal is emitted.

MBs are typically 25–35 nucleotides (nt) in length, with a specific probe sequence of 15–30 nt forming the loop, and a double-stranded stem of 5–8 base pairs [2]. MBs are more specific than conventional linear probes [3] and can distinguish between perfectly complementary targets and those with a single-nucleotide mismatch. In aqueous solutions, MBs hybridize to their targets more slowly than conventional linear probes [4] but may outperform them in organic solvents [5]. The hybridization rate is also dependent on the MB concentration, the accessibility of the target, the composition of the buffer, and other experimental conditions [6,7]. The fluorescence signal produced by MBs is proportional to the number of hybridized target

**Funding:** The author(s) received no specific funding for this work.

**Competing interests:** Oleh Fedorych is founder, shareholder and holds patent covering the method. Ruben R. Rosencrantz is a shareholder. Sophia Rosencrantz and Vitali Matyash have no competing interests to declare. This does not alter our adherence to PLOS ONE policies on sharing data and materials.

molecules therefore allowing quantitative analysis. The major drawback of the MB is their degradation [8,9] or spontaneous unfolding which results in grow of the MB fluorescence intensity even without hybridization. Even more, fluorescence intensity can be influenced by many environmental factors which significantly complicates its interpretation.

To overcome the limitations of MBs we propose a new method for the detection of MB hybridization that is based on the determination of the MB energy from fluorescence spectrum peak position. The peak position indicates an energy state of the MB, the hybridized state is red shifted relatively to the non-hybridized. The method offers the ability to distinguish between perfect duplexes and those with single mismatches.

## Materials and methods

Oligonucleotides were purchased from biomers.net (Ulm, Germany). The MB used in the experiments, comprises a 23-nt probe flanked by 6-bp tails (bold) to form a duplex stem 5′-**CGCGAT** TAGAGTTCCTGATCTTCTGGTCT **ATCGCG**-3′, labeled at the 5′-end with rhodamine 6G and at the 3′ end with the quencher BMN-Q620. Two complementary target strands were prepared, one of which was a perfect match (PM) for the probe 5′-CGCGAT AGACCAGA AGATCAGGAACTCTA ATCGCG-3′ whereas the other contained a mismatching (MM), central nucleotide (bold underlined), 5′-CGCGAT AGACCAGAA**A**ATCAGGAACTCTA ATCGCG-3′.

The MB was tested in four buffers. Lysis buffer 1 (LB1) was RAV-1 (Macherey-Nagel, Düren, Germany) similar to the buffer described by Boom and colleagues [10]. It contains ~5 M guanidinium thiocyanate, 50 mM Tris-HCl, 22 mM EDTA and 1.2% Triton X-100, and the pH is 7.0–7.5. Lysis buffer 2 (LB2) was prepared in-house and comprised 150 mM NaCl, 10 mM Tris-HCl (pH 7.4) and 0.25% Triton X-100 [11]. We also used 50 mM Tris-HCl (pH 7.4) and 100 mM HEPES-NaOH (pH 8.5) as non-lysing buffers. Cell lysate was prepared by solubilizing pellet from $5 \times 10^6$ cells of SLC-354 cell line (Hölzel Diagnostika Handels GmbH, Köln) in 200 μL LB1 buffer. Each liquid sample initially comprised 5 μL of 1 μM solution of the MB. Later into the samples we added 1 μL of the PM or MM targets (5 μM in Millipore water) or 1 μL of Millipore water as a control. Each sample was measured in accordance with the following routine. Fluorescence was measured twice for the sample containing the MB with a time gap between these measurements of at least 20 min. Immediately after the second measurement, we added one of the targets or water into the solution and the fluorescence was measured again. The fourth and the fifths measurements were made at least 10 min and at least 30 min after target addition, respectively. All measurements were taken at room temperature and all solutions were protected against ambient light.

The measurements were done using a custom-made micro-fluorescence device (Fig 1) in which the excitation light source was a 532-nm diode-pumped solid-state laser (5 mW emission power) passed through a 90:10 (Transmission:Reflection) beam splitter to give a final output of ~0.5 mW. This was reflected into the optical fiber using the coupler. The other end of the optical fiber (cut and polished) was immersed into the analyte. Part of the fluorescence emitted by the MB in the analyte couples back into the excitation fiber, and 90% of the fluorescence intensity after the beam splitter and high-pass fluorescence filter is coupled into the optical fiber using the second coupler, so that fluorescence was measured by the spectrometer. The wavelength scale of the spectra was converted to eV for convenience. The setup works in near-field configuration, so any changes in the refractive index of a test liquid (or related effects) can be considered negligible. The laser excitation power delivered to the working volume was adjusted to exclude or at least limit laser-induced heating.

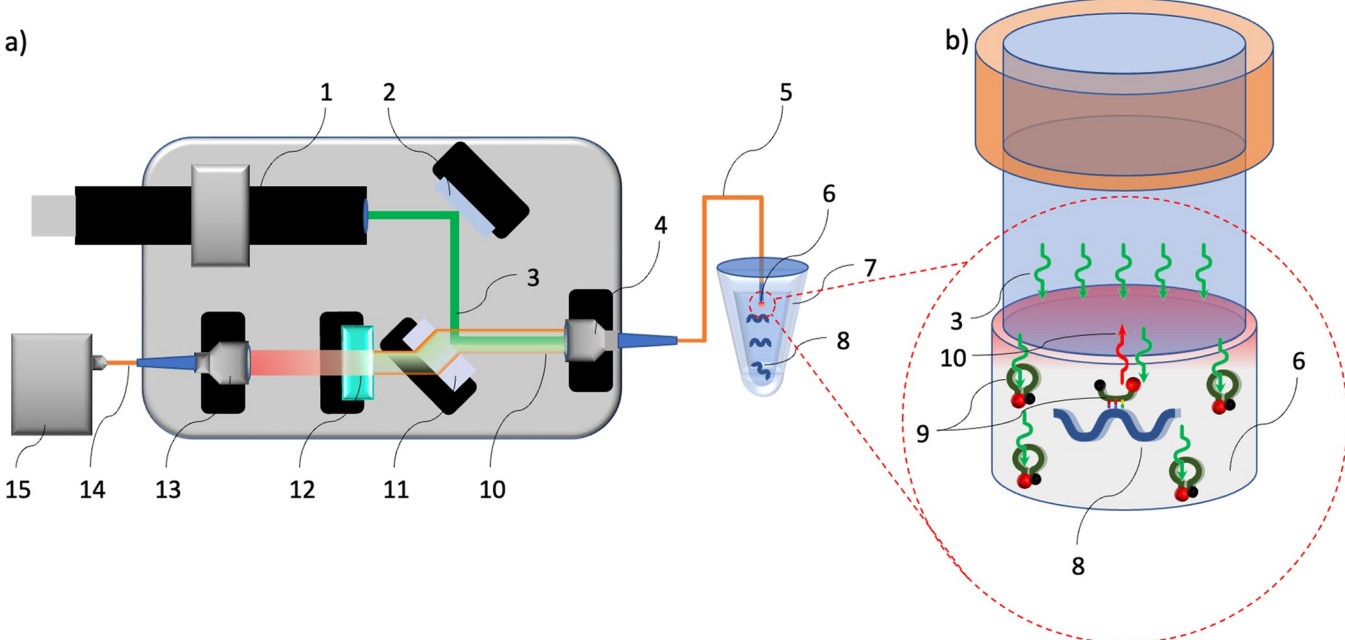

**Fig 1. The micro-fluorescence setup used to detect signals from the molecular beacon (MB) probes during hybridization.** (1) Excitation source: 532-nm laser. (2) Adjustment mirror. (3) Excitation beam. (4) Fiber coupler. (5) Optical fiber, the same fiber used to excite (3) the MB and for the collection (10) of emitted by MB light. (6) Working volume of the liquid (7), with target strands (8) that hybridize to the MB (9) thus causing it to emit light (10). The emitted light passes through the 90:10 beam splitter (11), fluorescence filter (12), fiber coupler (13) which couples the light in the fiber (14). The fluorescence is finally measured by a spectrometer (15).

## Results and discussion

The fluorescence signal changed when we added the PM complementary target strand to the MB probe, as shown by the difference between the solid and dashed lines in Fig 2A. For the HEPES buffer (red lines), adding the target strand to the MB reduced the fluorescence by more than 50%, which cannot be explained simply by the dilution effects when adding 1 μL of the target strand solution to 5 μL of the MB solution. For buffer LB2 (blue lines), adding the target strand to the MB probe had no effect on the signal. For the Tris-HCl buffer (yellow lines), the fluorescence signal doubled after adding the target strand. However, we observed a much more dramatic increase in the fluorescence signal when we added the target strand to the MB probe in buffer LB1 (black lines). Even without the target sequence, the fluorescence intensity was 2.5-fold higher in LB1 than HEPES and Tris-HCl, suggesting that many of the MBs existed in an unfolded state. Indeed, the fluorescence in LB1 without the target strand was stronger than the signal from the other three buffers in the presence of the target.

Fluorescence can be used for estimation of quantity of MBs emitting light. The number can be estimated from the fluorescence spectrum. First, the spectrum is integrated over all energies and the resulting value is recalculated in accordance with the experimental setup, considering each optical component: spectrometer sensitivity, quantum efficiency of the spectrometer grating, insertion losses (optical fiber, fiber coupler, fluorescence filter, beam splitter, optical coupler and optical fiber), and finally the coupling efficiency between the fiber and the analyte. The latter, also known as the numerical aperture, was used to estimate the portion of photons generated under excitation that might be coupled back into the fiber and thus determines the volume from which fluorescence is collected. Ray tracing simulations showed that not more than 15%, in best case, of all photons emitted may couple into the fiber, and the working volume may reach 60 nL. In the volume estimated amount of the MB is around ~$2\times10^6$ copies.

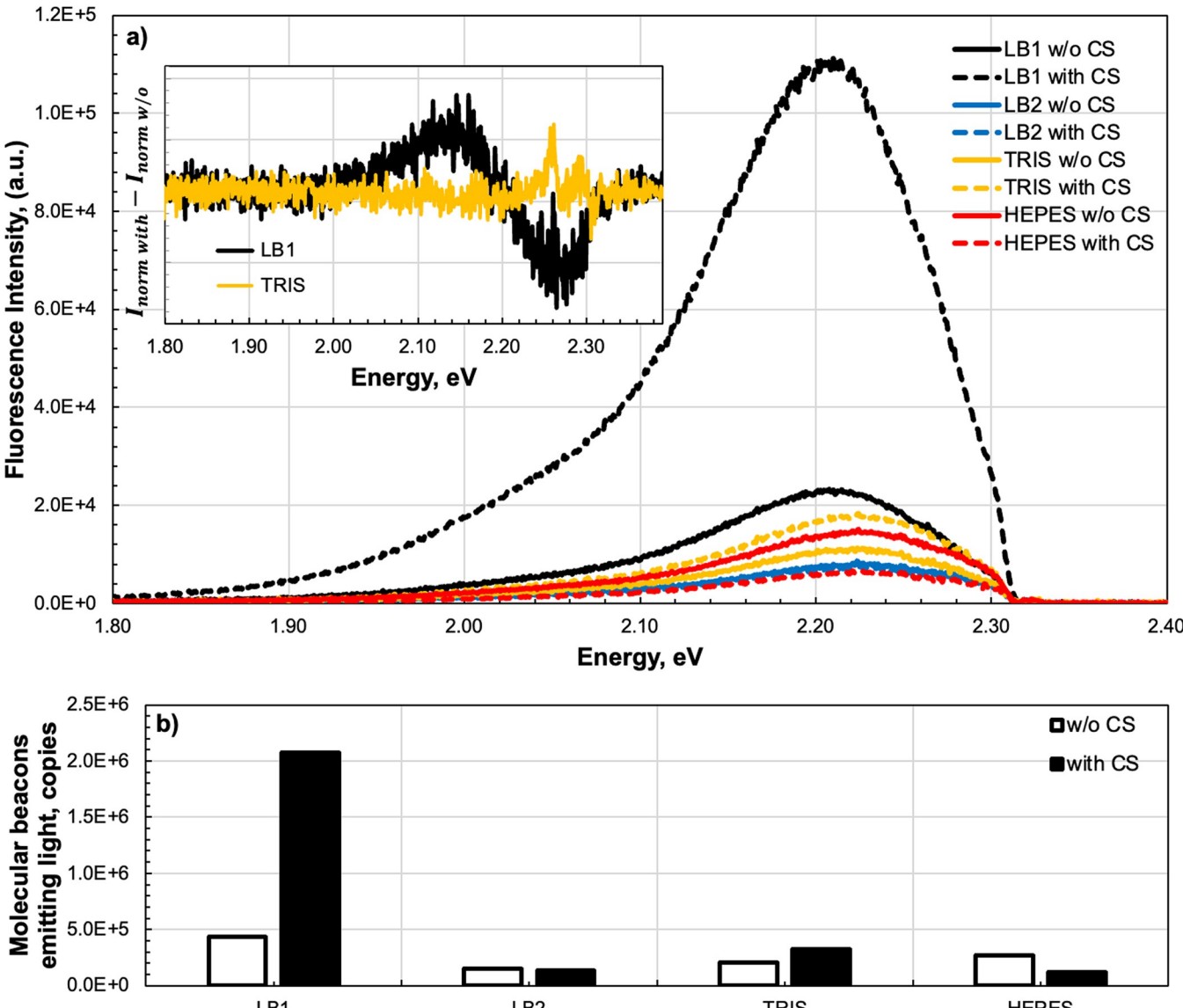

**Fig 2. Fluorescence signals detected using the micro-fluorescence setup.** (**a**) Molecular beacon (MB) fluorescence spectra acquired in buffers LB1, LB2, Tris-HCl and HEPES. The spectra were acquired without (solid lines) and with (dashed lines) the complementary strand (CS) in the buffer. Inset shows differences in the normalized intensities of the spectra with and without CS in buffers LB1 and Tris-HCl, the black line shape indicates that normalized spectra measured in presence of CS is red shifted. (**b**) Number of MB copies yielding fluorescence in buffers without CS (open bars) and 60 min after the addition of CS (black bars) for the respective spectra shown on the panel **a**).

Each unquenched rhodamine 6G molecule, under optical excitation, can emit only a certain number of photons [12]. This is the product of the dye's quantum efficiency (~95%) and inverse luminescent lifetime (~4 ns). The estimated numbers of MBs emitting photons (i.e., unfolded MBs) before and after adding the target strand are shown in Fig 2B. In LB2, Tris-HCl and HEPES, the estimates were ~$0.16 \times 10^6$, $0.2 \times 10^6$ and $0.26 \times 10^6$ MBs before adding the target strand, which is <10% of the total number of MBs in the working volume. Adding 1 µM of the target sequence therefore had the potential to increase the fluorescence signal by ~10-fold, but observed effect was much smaller ~$0.14 \times 10^6$, $0.32 \times 10^6$ and $0.12 \times 10^6$, respectively. However, in buffer LB1 the number of unfolded MBs was 2.5-times higher than in Tris-HCl ($0.38 \times 10^6$) and

increased by ~10-fold (almost $2 \times 10^6$) when the target strand was added. Comparing the number with estimated number of copies it can be concluded that nearly all MBs in the working volume were unfolded and emitting light, most likely due to complete hybridization. Comparing the amount of the MB in LB1 emitting light before and after adding the target strand, it is possible to estimate the MB melting temperature. For the MB in LB1 it should be not more than 4–5 K above the room temperature of 21˚C (294 K) and may thus reach 26˚C (297 K) versus 44˚C (317 K) MB melting temperature in Tris-HCl buffer.

The inset in Fig 2A shows the difference between normalized fluorescence intensities with and without the target strand in buffers LB1 and Tris-HCl. Whereas there was no change in Tris-HCl, the differential spectrum in buffer LB1 indicates that adding the target strand causes a redshift relative to the spectrum without the target strand, confirming that hybridization lowers the duplex free energy and this change is reported through the dye molecule attached to the MB. We therefore analyzed in more detail the effect of buffers with and without the target strand on the fluorescence energy of the MB, specifically its spectral position. All spectra were fitted with a double Gauss function, and the resulting fluorescence peak position (2.2–2.3 eV) and the corresponding peak amplitude, which is proportional to fluorescence intensity, were considered as indicators of MB energy and hybridization efficiency (Fig 3). Analyzing these two parameters, fluorescence peak position and peak amplitude, we can compare the method reported in the article with the fluorescence intensity typically used to assess MB hybridization [2]. The first two points of each dependency show the energies in buffers without the target strand. For Tris-HCl, the peak at 2.225 eV coincides precisely with the value specified for the dye, whereas it shifts to a higher value of 2.227 eV in HEPES and LB2. In contrast, the fluorescence peak in LB1 shifted to lower energies of 10 meV to 2.215 eV even without the target strand. Introducing the target strand did not influence the peak in Tris-HCl and increased the value by ~2 meV in HEPES and LB2. However, the emission position in LB1 was redshifted by about 4 meV from 2.215 eV to 2.211 eV, apparently due to the formation of a DNA duplex. This confirms the theoretically predicted [13] but until now unobserved decrease in MB free energy achieved by hybridization to a target sequence.

The emission properties of the rhodamine 6G are dependent on the surrounding liquid [14] (S1 Fig). To examine the impact of LB1 on the hybridization of the MB with the PM target strand and its influence on the fluorescence signal, we focused on the two substances that were not present in the other buffers: EDTA and guanidinium thiocyanate (Fig 3B and 3C). A comparison of 50 mM Tris-HCl with and without 22 mM EDTA revealed a negligible effect on fluorescence peak amplitude after adding the target strand, with no shift in the peak position. In contrast, a comparison of Tris-HCl with and without 5 M guanidinium thiocyanate showed that the buffer with guanidinium thiocyanate behaved similarly to LB1. The initial fluorescence peak amplitude without the target strand was at least twice that of the Tris-HCl buffers with and without 22 mM EDTA. The spectral position of the fluorescence peak was ~2.215 eV, and the addition of the target increased the peak amplitude and caused a redshift by ~4 meV. This confirmed that the behavior of LB1 is determined mainly by the presence of the chaotropic agent guanidinium thiocyanate. The red shift of fluorescence peak was observed for other MB's hybridizing with PM complementary targets (S2 and S3 Figs), however the shift was not observed for the MB with non-complementary targets (S4 Fig).

One of the criteria determining the rate of MB hybridization is the availability of unfolded MB molecules [7]. The MB must open its stem before hybridization, which means at least the same energy that is achieved by its folding must be applied to the folded MB. A lack of unfolded MBs slows down the kinetics or even aborts the hybridization process, which is supposed to be more favorable for the MB than staying folded. For example, the loss of fluorescence intensity in HEPES buffer after the target strand was added (increasing the emission

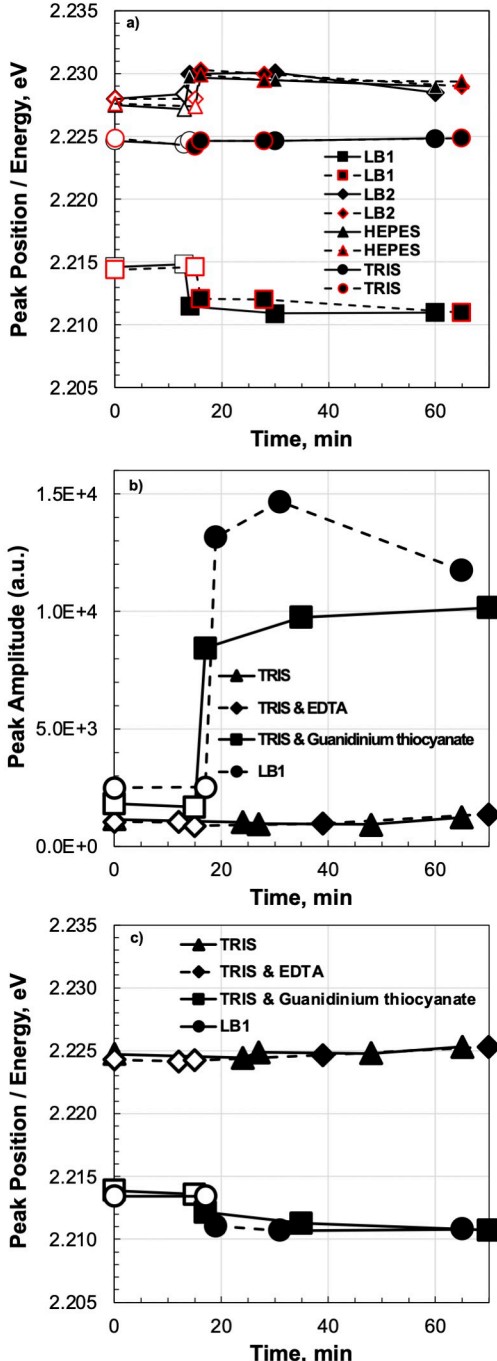

**Fig 3. Characteristics of the fluorescence peaks.** (**a**) Peak fluorescence signals measured for molecular beacons (MBs) diluted in four different buffers (identified in the key). Open dots correspond to the MB state without a complementary target, and closed dots represent the state with a complementary target. The first measurement was taken less 1 min after adding the target. (**b**, **c**) Comparison of buffers to identify components responsible for the unique behavior of MBs in buffer LB1. (**b**) Fluorescence peak amplitude and (**c**) fluorescence peak position for MBs with and without the complementary target strand (added ~15 min after the beginning of sample analysis). Each data point is accompanied by its corresponding standard deviation bars.

position by ~2 meV) may indicate that the internal energy of the MB increased, but not enough to make hybridization possible. This may explain the 2 meV blueshift of the MB in these buffers compared to Tris-HCl, even before the target was added, in contrast to the slow increase in fluorescence intensity in Tris-HCl. These data suggest that the energy barrier to open a MB in these buffers could reach ~4 meV or ~46 K. Moreover, the fluorescence observed in these buffers could be attributed to misfolded MBs, with the fluorescence intensity changes resulting from structural reorganization of the MBs (reducing the number of misfolded MBs in HEPES buffer but remaining the same in LB2). In LB1, the fluorescence in the absence of the target strand was 2.5-fold higher than in HEPES buffer, possibly because up to half of the fluorescence signal intensity in LB1 can be attributed to already unfolded MBs. Adding the target strand triggers the hybridization of these unfolded MBs and their transition to lower energy levels, with simultaneous unfolding of further MBs until they are all hybridized, or they reach equilibrium. In accordance with this observation, MBs gain at least ~3.6 meV or ~42 K from hybridization.

To verify the results, we tested the influence of $Na^+$ and $Mg^{2+}$ at different concentrations in buffer LB1 on the MB emission peak before and after adding the target strand (Fig 4A and 4B). Theoretical dependence of the MB relative melting temperature and the MB/target duplex melting temperature [15] on $Na^+$ and $Mg^{2+}$ concentration was calculated using the UNAfold [16] and the nearest neighbor model [13]. In experimental data, the energy states and transitions of the MB are reported relatively to main energy state of rhodamine 6G dye (S1 Fig). Moreover, the absolute energy of the folded MB ($E_0$) is unknown and cannot be measured due to quenching of its fluorescence, but it can be determined based on the experiment. Assuming the fluorescence peak position in presence of the target corresponds to energy of MB/target duplex then the $E_0$ level must be above of the duplex energy exactly by the value of hybridization energy ($E_H$). The best match was obtained at $E_0 = 2.2125$ eV. Also, theoretical values of MB folding/unfolding energy ($E'_U$) are higher than measured (black line on Fig 4A and 4B). A match occurs if the value is reduced by 2.5-fold due to effect of guanidinium thiocyanate on the MB melting temperature [17], so $E_U = E'_U/2.5$ and the fluorescence energy change is $\Delta E_F = E_U + E_H$ perfectly matching the experimental data (red dashed line).

We constructed an energy diagram to illustrate the process of hybridization in a simpler manner, where the ground state of the unfolded MB is fixed. All other states are shown relatively to it (Fig 4C). This revealed that shallow energy minima for the folded MB facilitate unfolding and hybridization with available targets where the state with minimal possible energy to be formed. For example, the barrier to unfold MBs in buffer LB1 is ~2.35 meV or ~27 K (agree with the melting temperature estimated above), compared to ~3.7 meV or ~43 K in HEPES. As it is seen, the shallow energy minima are in the range of the experiment thermal energy. The shallowness of the levels leads to spontaneous MB folding/unfolding and results in the fluctuations of fluorescence intensity. This complicates application of the fluorescence intensity for detection of the hybridization. The line position, which exactly reflects the MB state is the only accurate indicator of hybridization. The energy transition between unfolded and hybridized states, reported herein, should not be confused with dye fluorescence or fluorescence resonance energy transfer (FRET). It is additional to FRET and occurs only when a fluorophore-labeled molecule forms a bond with another molecule. The energy diagram also helps to understand why the effect is not observed in the absence of guanidinium thiocyanate. The main reason is absence of unfolded MB which might be accessed by the targets.

We also performed experiments where together with the complementary target strands in LB1 were supplemented with lysed cells, see Fig 5A and 5B). The MB duplex state cannot be clearly discriminated from fluorescence intensity Fig 5A). This result was expected, as adding of the cells into the reaction might change the optical properties of the LB1, specifically its

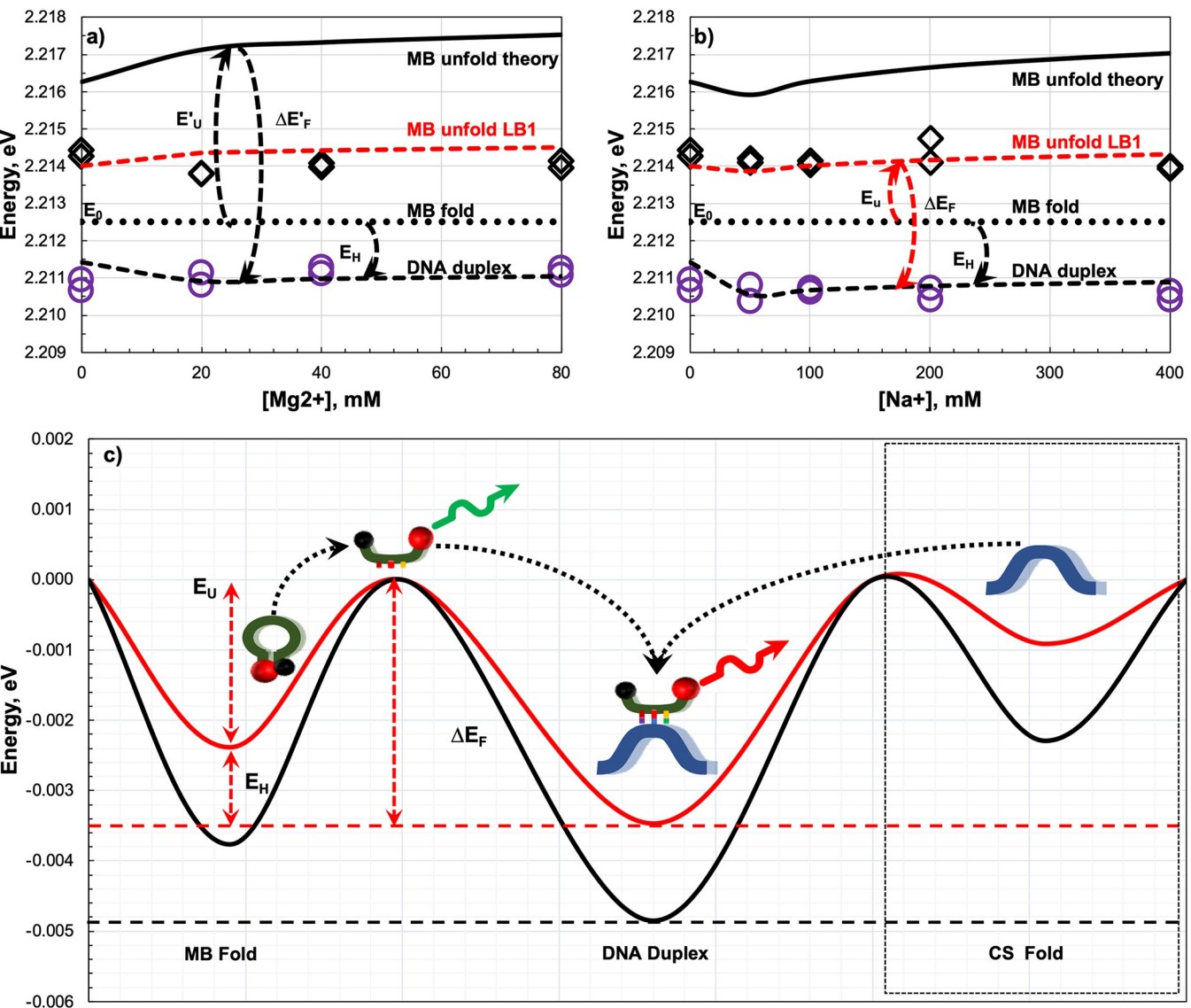

**Fig 4. Fluorescence peak position as a function of Mg²⁺ and Na⁺ concentrations in buffer LB1.** (**a**) Fluorescence peak position as a function of the $Mg^{2+}$ concentration. (**b**) Fluorescence peak position as a function of the $Na^+$ concentration. Open black diamonds show the fluorescence peak position in LB1 without a target strand. Open violet dots show the peak following the addition of the target strand. Black solid line shows the theoretical dependence of the MB melting (unfolding) energy on the concentration of $Mg^{2+}$ and $Na^+$. Red dashed line shows the expected MB melting energy. Black dashed line shows the expected MB/target duplex energy. Black dashed arrows show transitions expected from the theory and red dashed arrows show those observed in the experiment. $E'_U$ and $E_U$ are the theoretical and empirical MB unfolding energies. $\Delta E'_F$ and $\Delta E_F$ are the theoretical and empirical shifts in fluorescence energy due to hybridization. $E_H$ is the gain in MB free energy due to hybridization and $E_0$ is the arbitrary MB free energy in the liquid. (**c**) Energy diagram of MB/target duplexes in HEPES (black line) or LB1 (red line). The energy of the unfolded MB was set to zero and other states are shown relative to this. The relative height of the energy barriers is estimated from the MB and MB/target duplex melting temperatures. Dotted arrows show structural transformations of the MB during hybridization.

turbidity. Additionally, the cell lysate also may influence the MB folding/unfolding resulting in changes of measured fluorescence intensity. An effect of the cell lysate on the MB hybridization energy was not observed Fig 5B), the samples with and without complementary target are well resolved. As it seen from the Fig 3B and 3C, S2–S4 Figs. the fluorescence intensity and MB hybridization energy shows similar characteristics, however the experiment with the lysed

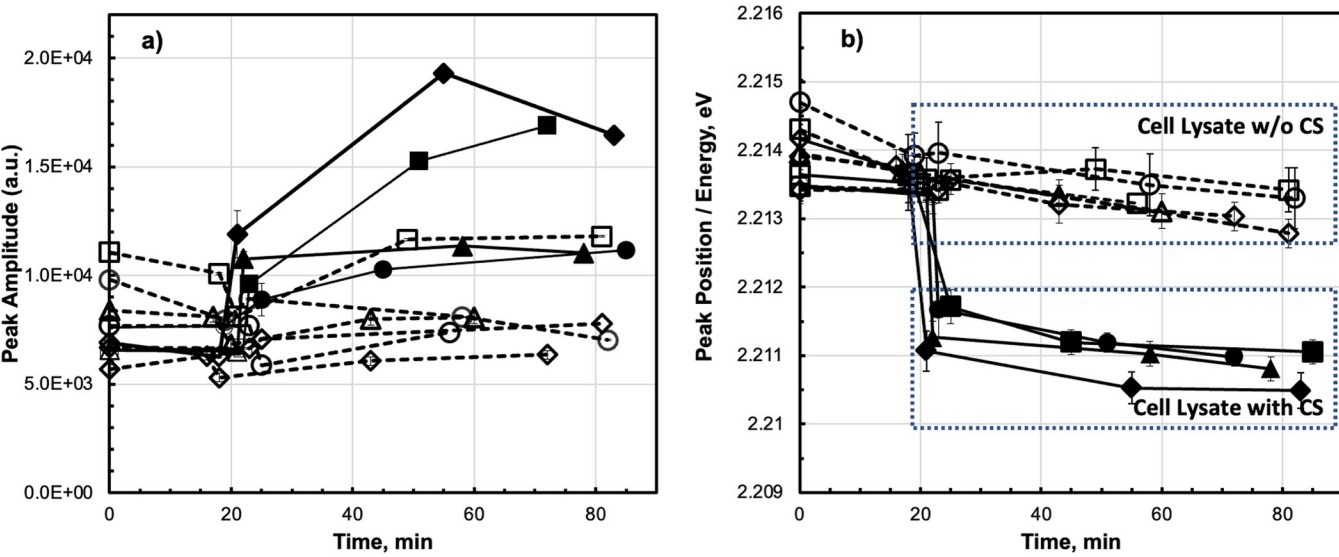

**Fig 5.** Time course of fluorescence peak amplitude (**a**) and peak position (**b**) of the molecular beacon in LB1. Immediately after the second measurement (point) cell lysate with (closed points) or without (open points) complementary target strand was added. Each data point is accompanied by its corresponding standard deviation bars.

cells reconfirms the above conclusion that the MB energy is more valid hybridization benchmark.

To test the method limits, we performed experiment where hybridization of the perfectly matching stand and strand with a single-nucleotide mismatch was done (Fig 6). The measurements as before were performed in the LB1. The result we obtained was completely unexpected. In accordance with the theory, the MB will hybridize with the MM and PM strands, although the MM duplex state must be on ~0.3 meV higher in energy as the duplex PM state. Distinction between PM and MM was not expected and should not be measured at least on the available hardware. Nevertheless, crystal clear differentiation between DNA duplexes formed with the PM and MM target strands is observed in Fig 6. Even more, the energy of the MM duplex, was $\Delta\Delta G$ = -0.9 meV (-0.02 kcal/mol) lower than the energy of the PM duplex, indicating that the less stable MM state become energetically favorable as the most stable PM state. The possibility that the MM state may occupy lower energies as the PM state was already theoretically predicted [18–21], where changes of the free-energies of the PM and MM states are considered. The free-energy difference between these states is small due to enthalpy-entropy compensation [22] observed in aqueous solutions. Exclusion of the water and substitution it with the 5M guanidinium thiocyanate as well as the free hydrogen bond in the C•A non-matching pair leads to favoring of the state over the DNA duplex formed with perfectly matching base pairs. Although this effect should be studied in detail, it helped us to resolve the PM and MM states and may have critical value for understanding the nature of molecular interactions resulting in DNA/RNA mutations.

## Conclusions

The new method of RNA/DNA probes hybridization study was proposed. Inclusion of chaotropic agents allows rapid and specific hybridization even at room temperature, where hybridization can directly be observed by the reduction of free energy of the duplex states. Moreover, the method can distinguish between hybridization events involving perfectly complementary strands and those containing a single mismatch, where the central role may involve enthalpy-

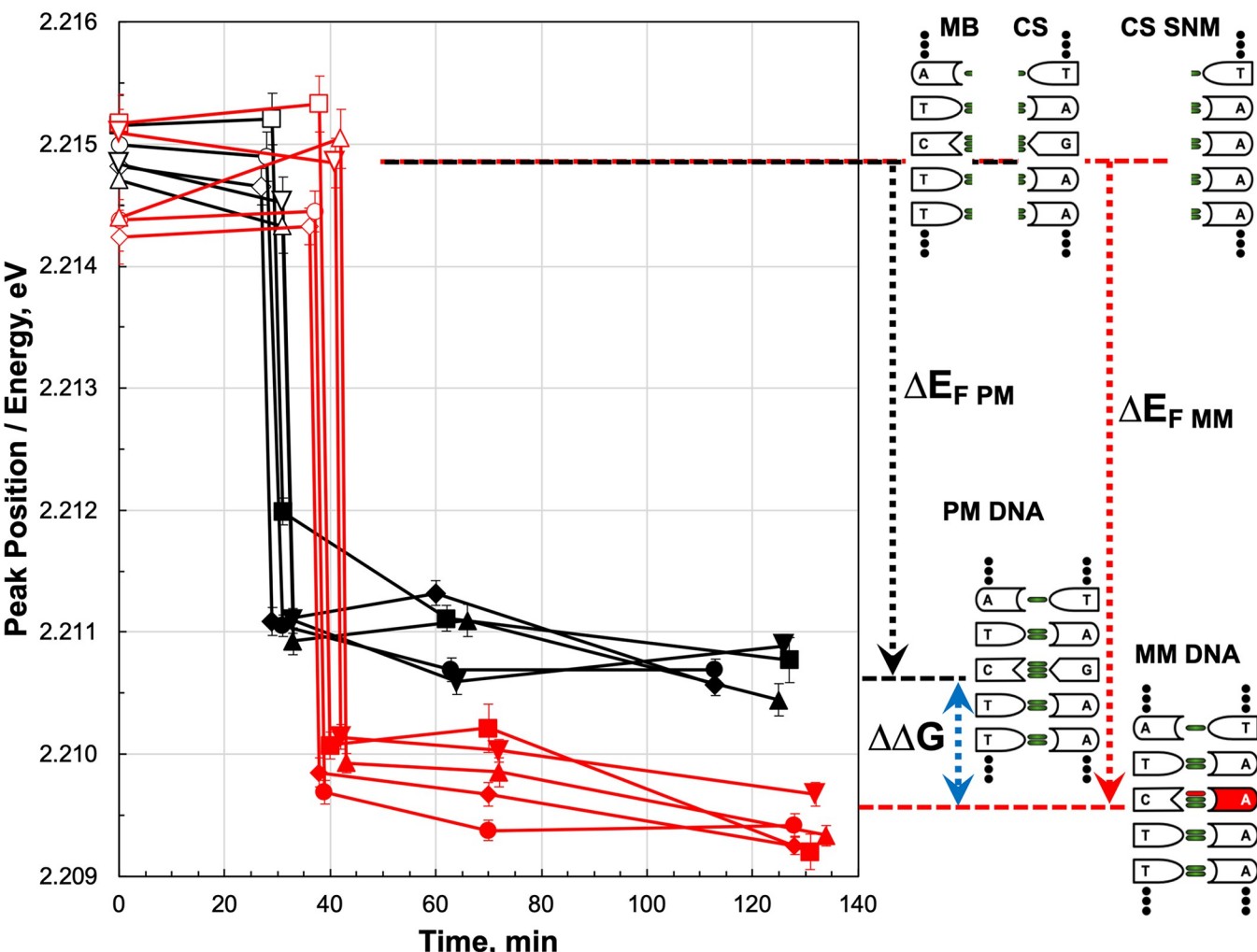

**Fig 6. Fluorescence peak position for the molecular beacon (MB) in buffer LB1.** Measurements are shown before (first two points) and after (third point onwards) the formation of a duplex with a perfectly matching target (PM, black) and a target containing a single mismatch (MM, red). The five base pairs surrounding the position of the MM are shown for the PM and MM targets on the right, with green representing hydrogen bonds and red a missing hydrogen bond. ΔΔG is the free energy difference between the duplexes formed with the PM and MM strands. CS- complementary strand and CS SNM—complementary strand with single nucleotide mismatch.

entropy compensation. Our findings match the theoretical model and offer a better understanding of the hybridization between RNA/DNA probes and their target strands.

## Supporting information

**S1 Fig. Emission spectrum of rhodamine 6G in three buffers.** The buffers are Tris-HCl (black), HEPES (blue), LB1 (containing guanidinium thiocyanate) (red) and LB2 (containing Triton X-100) (green).
(TIF)

**S2 Fig.** Fluorescence peak amplitude (a) and peak position (b) of the molecular beacon 5'-
CGCGATC ATTACTTATAGGGATGGCTATC GATCGCG-3' in LB1 without (open points and dashed lines) and with complementary strand (solid points and solid lines). Each data point is accompanied by its corresponding standard deviation bars.
(TIF)

**S3 Fig.** Fluorescence peak amplitude (a) and peak position (b) of the molecular beacon 5'-`CG CGATC AAATGCCAGTGTTATCC GATCGCG`-3' in LB1 without (open points and dashed lines) and with complementary strand (solid points and solid lines). Each data point is accompanied by its corresponding standard deviation bars.
(TIF)

**S4 Fig.** Fluorescence peak amplitude (a) and peak position (b) of the molecular beacon 5'-`CG CGAT TAGAGTTCCTGATCTTCTGGTCT ATCGCG`-3' in LB1 without (open points) and with (solid points) target strand. Red color corresponds to the samples with complementary strand (CS). Green, dark blue and violet color to the samples with non-complementary strands (NCS). Each data point is accompanied by its corresponding standard deviation bars.
(TIF)

**S1 File. Recommended procedure to estimate number of dye molecules emitting light.**
(DOCX)

## Acknowledgments

We thank biomers.net and Dr. Barbara Pohl, particularly for help with of the MBs design.

## Author Contributions

**Conceptualization:** Sophia Rosencrantz, Oleh Fedorych.

**Investigation:** Sophia Rosencrantz, Vitali Matyash, Oleh Fedorych.

**Methodology:** Sophia Rosencrantz, Vitali Matyash, Ruben R. Rosencrantz, Oleh Fedorych.

**Visualization:** Oleh Fedorych.

**Writing – original draft:** Oleh Fedorych.

**Writing – review & editing:** Sophia Rosencrantz, Vitali Matyash, Ruben R. Rosencrantz, Oleh Fedorych.

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
