## [Decision Letter · Decision Letter 0]

28 May 2024

PONE-D-24-18169A method to measure molecular hybridization.PLOS ONE

Dear Dr. Fedorych,

Thank you for submitting your manuscript to PLOS ONE. After careful consideration, we feel that it has merit but does not fully meet PLOS ONE’s publication criteria as it currently stands. Therefore, we invite you to submit a revised version of the manuscript that addresses the points raised during the review process.

We look forward to receiving your revised manuscript.

Kind regards,

Yu-Hsuan Tsai

Academic Editor

PLOS ONE

Journal Requirements:

"The authors declare the following interests: Oleh Fedorych is founder, shareholder and holds patent covering the method. Ruben R. Rosencrantz is a shareholder. Sophia Rosencrantz and Vitali Matyash have no competing interests to declare."

4. We notice that your supplementary figures are uploaded with the file type 'Figure'. Please amend the file type to 'Supporting Information'. Please ensure that each Supporting Information file has a legend listed in the manuscript after the references list.

Reviewers' comments:

Reviewer's Responses to Questions

**Comments to the Author**

1. Is the manuscript technically sound, and do the data support the conclusions?

Reviewer #1: Partly

Reviewer #2: Partly

2. Has the statistical analysis been performed appropriately and rigorously? 

Reviewer #1: N/A

Reviewer #2: Yes

3. Have the authors made all data underlying the findings in their manuscript fully available?

Reviewer #1: Yes

Reviewer #2: Yes

4. Is the manuscript presented in an intelligible fashion and written in standard English?

Reviewer #1: Yes

Reviewer #2: Yes

5. Review Comments to the Author

Reviewer #1: The manuscript tried to develop a new method for the quantitation of MB hybridization based on the measurement of changes in free energy. It could be an interesting work. However, the manuscript is not well-organized, and also the novelty is not clear, and the advantage of the established method has also not been proved. Based on fluorescence intensity, we can easily check the status of MB hybridization. Moreover, all the figures are not clear. The method validation is also not prepared.

Reviewer #2: The authors present a novel method to quantify MB hybridization by measuring changes in free energy via micro-fluorescence detection. This approach allows direct determination of hybridization energy from fluorescence spectra and distinguishes between the MB's unfolded and hybridized states. Additionally, it may differentiate between perfectly complementary DNA duplexes and those with a single-nucleotide mismatch.

1. In Fig. S1, why don’t you show the spectrum of Rh6G in Tris buffer? It would be important to compare the effects of all four buffers.

2. In materials and methods parts, you mentioned that you measure spectrum several times in various time points. For spectrums shown in Fig. 1 and S1, which time point are those spectrums from?

3. Could you explain why the guanidinium thiocyanate in LB1 has such impacts on the hybridization of the MB with the PM target strand? Is there any literature studying the effects of molecules or similar molecules (other detergents) in hybridization before? It’s hard for me to believe no body did those basic research (solvent effect on hybridization) before.

4. In Fig 6, the results are very unexpected. As your theory cannot explain this phenomenon, it would be not appropriate to claim that “the method enabled us to discriminate between DNA duplexes with perfect complementarity or a single-nucleotide mismatch”. You may use “probably”.

5. In Fig 6, have you ever done the experiment in the other 3 buffers? These experiments may give you more clues.

6. In Fig 3a, the symbols are quite hard to understand. Please make them more clear.

6. PLOS authors have the option to publish the peer review history of their article (what does this mean?). If published, this will include your full peer review and any attached files.

Reviewer #1: No

Reviewer #2: **Yes: **Zefan Li

---

## [Author Response · Author response to Decision Letter 0]

11 Jul 2024

Reviewer's comments to the author:

Reviewer #1: The manuscript tried to develop a new method for the quantitation of MB hybridization based on the measurement of changes in free energy. 

Author response: Thank you, however it must be pointed out that the method presented is not quantitative. Indeed, in the manuscript we have made the attempt to estimate number of the molecular beacons in unfolded state (emitting light) in various buffers before and after adding of control strand. However, the message we would like to deliver, that destabilization of the stem by reduction of the MB melting temperature allows to open certain amount of the MB, make them available for hybridization with the complementary target strands. Consequently, the large number of unfolded MB (~25%) enables fast (almost instantly) and complete (nearly ~100%) hybridization of the MB with the control strands. 

Reviewer #1: It could be an interesting work. However, the manuscript is not well-organized, and also the novelty is not clear, and the advantage of the established method has also not been proved. Based on fluorescence intensity, we can easily check the status of MB hybridization. Moreover, all the figures are not clear. The method validation is also not prepared.

Author response: We appreciate the Reviewer #1 feedback and would like to thank to the Reviewer #1 for pointing out the omission in the manuscript. Mainly, we haven't stated that the "Peak Amplitude" parameter is proportional to fluorescence intensity.

We would like to amend the manuscript on the page 7, line 176 add the following:

 , which is proportional to fluorescence intensity, 

and on the page 7 line 177 made the following amendment:

Analyzing these two parameters, fluorescence peak position and peak amplitude, we can compare the method reported in the article with the fluorescence intensity typically used to assess MB hybridization (2).

Fig.3b, Fig. 5a, S2a Fig., S3a Fig., and S4a Fig shows comparison between the peak amplitude, or as it is just explained the value proportional to "fluorescence intensity", and the energy of the MB's, see Fig.3c, Fig. 5b, S2b Fig., S3b Fig., and S4b Fig. In the manuscript already shown results of measurements of 3 different sequences of the MBs. As it can be seen from the Fig's, in most cases the approach proposed in the manuscript and the fluorescence-based approach working quite well. For the hybridization performed in presence of the lysed cells Fig. 5a the intensity-based method doesn't allow to distinguish between the samples where MB have hybridized with the target, fluctuations of the peak amplitude (fluorescence intensity) are too strong. Although, the energy-based approach, proposed in the manuscript, allows clearly distinguish between the MB in unfolded and hybridized states even with presence of the lysate. In the manuscript we have cited only two examples, please see references (8, 9), where MB fluorescence intensity provides unreliable results. In the literature can be found much more examples when the fluorescence-based approach producing erroneous result. Solving the problem with application of the different approach, at least deserve attention of the scientific community.

To emphasize the above, we would like to introduce the following amendment on the page 11 line 293.

As it seen from the Fig. 3b) and Fig. 3c), S2 Fig., S3 Fig. and S4 Fig. the fluorescence intensity and MB hybridization energy shows similar characteristics, however the experiment with the lysed cells reconfirms the above conclusion that the MB energy is more valid hybridization benchmark.

We also would like to underline that the "Peak Amplitude" is more accurate parameter as that "Fluorescence Intensity". The peak amplitude corresponds to the intensity of light emitted exclusively by the fluorescent dye molecules only in the monomer state (Penzkofer A, Leupacher W. Fluorescence behaviour of highly concentrated rhodamine 6G solutions. Journal of luminescence. 1987 May 1;37(2):61-72.). Fluorescent intensity-based measurements, used to characterize the MB hybridization, operating with integral intensity which comprises light emitted by the dye in the mono- and in the dimer states. Here, someone can expect measurement error of about 10-20% of the total intensity.

Reviewer #2: The authors present a novel method to quantify MB hybridization by measuring changes in free energy via micro-fluorescence detection. This approach allows direct determination of hybridization energy from fluorescence spectra and distinguishes between the MB's unfolded and hybridized states. Additionally, it may differentiate between perfectly complementary DNA duplexes and those with a single-nucleotide mismatch.

1. In Fig. S1, why don’t you show the spectrum of Rh6G in Tris buffer? It would be important to compare the effects of all four buffers.

Author response: Thank you for the suggestion, normalized spectrum of Rh6G fluorescence is added to the S1. Fig (black line), caption of the S1. Fig also changed. On page 17, line 441 added amendment:

 Tris-HCl (black)

2. In materials and methods parts, you mentioned that you measure spectrum several times in various time points. For spectrums shown in Fig. 1 and S1, which time point are those spectrums from?

Author response: Probably the Reviewer meant Fig 2a). Fig. 1 shows the experimental setup. If our assumption is correct, then the spectra with and without complementary target strand shown in the Fig 2a) are taken just before and immediately after adding of the complementary strand. In absolute units, the time interval is at least two minutes and can reach five minutes in the worst case.

3. Could you explain why the guanidinium thiocyanate in LB1 has such impacts on the hybridization of the MB with the PM target strand? Is there any literature studying the effects of molecules or similar molecules (other detergents) in hybridization before? It’s hard for me to believe no body did those basic research (solvent effect on hybridization) before.

Author response: In the manuscript we analyzed two aspects of MB hybridization and with PM target strand:

First, peak amplitude, which is proportional to fluorescence intensity, which is a standard measurement parameter. The fundamental study of the influence of various solvents on MB hybridization is made in the reference (5). Hybridization of RNA in solutions containing guanidinium thiocyanate was studied in reference (17). It is known, that solvents, besides significant acceleration of the hybridization, also increases background fluorescence, caused by spontaneous unfolding of the MB. This limits application of the solvents in fluorescence-intensity measurements because discrimination between the spontaneously unfolded MB and the MB hybridized with a target strand is impossible.

Second, peak position reports the relative energy of the MB in the buffers. Here the role of the guanidinium thiocyanate is destabilization the MB's stem and unfolding the MB's. It changes fluorescence of the Rhodamine 6G but we would not relate the change of the peak position for the MB-PM target strand duplex with the guanidinium thiocyanate. We have observed similar behavior (decrease of the energy in result of nucleic acids hybridization) in other buffers and for other structures too.

4. In Fig 6, the results are very unexpected. As your theory cannot explain this phenomenon, it would be not appropriate to claim that “the method enabled us to discriminate between DNA duplexes with perfect complementarity or a single-nucleotide mismatch”. You may use “probably”.

Author response: Indeed, the results in the Fig.6 were unexpected. We were expecting that the MM duplex to be above of the PM duplex energy also we were not expecting resolution of these two states. 

We would like to thank to the Reviewer #2 for pointing out that our explanation of the phenomenon shown on Fig.6 needs improvement. 

The theory explaining the effect was developed decades ago. The nearest neighbor (NN) theory referred in the manuscript (13), cannot explain the effect because it assumes that enthalpy and entropy are linearly correlated, and hybridization occurs in aqueous solutions. Exclusion of water and substitution it with a "solvent", such as or similar to 5M guanidinium thiocyanate, may lead to the situation observed. It becomes possible only when the equilibrium state of the DNA duplex is perturbated with the broken hydrogen bond (please see references 18-21 in the revised version of the manuscript). Therefore, we would like to keep the claim "the method enabled us to discriminate between DNA duplexes with perfect complementarity or a single-nucleotide mismatch". However, we propose the following amendment on the page 12 lines 314-320:

This contradicts the nearest neighbor model and all other models currently used to describe RNA/DNA hybridization. An additional mechanism must be considered. It can be enthalpy–entropy compensation mechanism (18), where small changes in entropy are not compensated by enthalpy in terms of changes in free energy (19). To our knowledge, the enthalpy–entropy compensation mechanism effect has not been directly observed before. Key parameters that increase the entropy of the relatively unstable duplex include the free hydrogen bond in the C•A non-matching pair and the chaotropic agent as a solvent in the aqueous solution, favoring this state over the stable DNA duplex with perfectly matching base pairs.

to be replaced:

The possibility that the MM state may occupy lower energies as the PM state was already theoretically predicted (18-21), where changes of the free-energies of the PM and MM states are considered. The free-energy difference between these states is small due to enthalpy-entropy compensation (22) observed in aqueous solutions. Exclusion of the water and substitution it with the 5M guanidinium thiocyanate as well as the free hydrogen bond in the C•A non-matching pair leads to favoring of the state over the DNA duplex formed with perfectly matching base pairs.

the following references with respective indexes are added:

18. Petruska J, Sowers LC, Goodman MF. Comparison of nucleotide interactions in water, proteins, and vacuum: model for DNA polymerase fidelity. Proceedings of the National Academy of Sciences. 1986 Mar;83(6):1559-62.

19. Petruska J, Goodman MF, Boosalis MS, Sowers LC, Cheong C, Tinoco Jr I. Comparison between DNA melting thermodynamics and DNA polymerase fidelity. Proceedings of the National Academy of Sciences. 1988 Sep;85(17):6252-6.

21. Goodman MF. Hydrogen bonding revisited: geometric selection as a principal determinant of DNA replication fidelity. Proceedings of the National Academy of Sciences. 1997 Sep 30;94(20):10493-5.

the following indexes are changed:

the reference 18,

18. Lumry R, Rajender S. Enthalpy–entropy compensation phenomena in water solutions of proteins and small molecules: a ubiquitous properly of water. Biopolymers: Original Research on Biomolecules. 1970 Oct;9(10):1125-227.

become new index 22:

22. Lumry R, Rajender S. Enthalpy–entropy compensation phenomena in water solutions of proteins and small molecules: a ubiquitous properly of water. Biopolymers: Original Research on Biomolecules. 1970 Oct;9(10):1125-227.

The reference 19,

19. Petruska J, Goodman MF. Enthalpy-Entropy Compensation in DNA Melting Thermodynamics (∗). Journal of Biological Chemistry. 1995 Jan 13;270(2):746-50.

become index 20:

20. Petruska J, Goodman MF. Enthalpy-Entropy Compensation in DNA Melting Thermodynamics (∗). Journal of Biological Chemistry. 1995 Jan 13;270(2):746-50.

5. In Fig 6, have you ever done the experiment in the other 3 buffers? These experiments may give you more clues.

Author response: We tested hybridization of the MB with complementary target strands in other buffers too. The MB at room temperature has shown extremely slow hybridization dynamic, due to very good stem stability, as it is written on the page 11 line 283:

The energy diagram also helps to understand why the effect is not observed in the absence of guanidinium thiocyanate. The main reason is absence of unfolded MB which is accessed by the targets.

This complicates explicit comparison between the LB1 and the other 3 buffers. However, our unpublished preliminary results measured for other oligonucleotides in water-based buffers (not published yet), without any solvents, confirms the theoretical assumption that the effect shown in Fig.6 is the result of water exclusion and substitution it with the solvent.

6. In Fig 3a, the symbols are quite hard to understand. Please make them more clear.

Author response: Dear Reviewer #2 thank you for the remark. To improve visual acceptance and resolution of the overlapping points each second sample in the duplicates now is shown with red border line. Please see the Fig. 3a revised.

---

## [Decision Letter · Decision Letter 1]

17 Jul 2024

A method to measure molecular hybridization.

PONE-D-24-18169R1

Dear Dr. Fedorych,

We’re pleased to inform you that your manuscript has been judged scientifically suitable for publication and will be formally accepted for publication once it meets all outstanding technical requirements.

Kind regards,

Yu-Hsuan Tsai

Academic Editor

PLOS ONE

Additional Editor Comments (optional):

Reviewers' comments:

Reviewer's Responses to Questions

**Comments to the Author**

1. If the authors have adequately addressed your comments raised in a previous round of review and you feel that this manuscript is now acceptable for publication, you may indicate that here to bypass the “Comments to the Author” section, enter your conflict of interest statement in the “Confidential to Editor” section, and submit your "Accept" recommendation.

Reviewer #1: All comments have been addressed

Reviewer #2: All comments have been addressed

2. Is the manuscript technically sound, and do the data support the conclusions?

Reviewer #1: Yes

Reviewer #2: Yes

3. Has the statistical analysis been performed appropriately and rigorously? 

Reviewer #1: Yes

Reviewer #2: Yes

4. Have the authors made all data underlying the findings in their manuscript fully available?

Reviewer #1: Yes

Reviewer #2: Yes

5. Is the manuscript presented in an intelligible fashion and written in standard English?

Reviewer #1: Yes

Reviewer #2: Yes

6. Review Comments to the Author

Reviewer #1: All the comments have been addressed, and the current one can be accepted for publication in PLOS ONE.

Reviewer #2: The author has responded to all my questions. I think the article can be accpeted. I'm still wondering if the phenomenon in the Fig.6 would happen in other DNA or only in this specific DNA.

7. PLOS authors have the option to publish the peer review history of their article (what does this mean?). If published, this will include your full peer review and any attached files.

Reviewer #1: No

Reviewer #2: No

---

## [Editor Report · Acceptance letter]

8 Aug 2024

PONE-D-24-18169R1 

PLOS ONE

Dear Dr. Fedorych, 

I'm pleased to inform you that your manuscript has been deemed suitable for publication in PLOS ONE. Congratulations! Your manuscript is now being handed over to our production team.

Kind regards, 

on behalf of

Dr. Yu-Hsuan Tsai 

Academic Editor

PLOS ONE